# Lipases secreted by a gut bacterium inhibit arbovirus transmission in mosquitoes

**Xi Yu** [1,2,3,4☯], **Liangqin Tong** [1,2,3☯], **Liming Zhang** [1,2,3☯], **Yun Yang** [1,2,3], **Xiaoping Xiao** [1,3], **Yibin Zhu** [1,2,3,5], **Penghua Wang** [6], **Gong Cheng** [1,2,3]*

**1** Tsinghua-Peking Center for Life Sciences, School of Medicine, Tsinghua University, Beijing, China, **2** Institute of Infectious Diseases, Shenzhen Bay Laboratory, Shenzhen, Guangdong, China, **3** Institute of Pathogenic Organisms, Shenzhen Center for Disease Control and Prevention, Shenzhen, Guangdong, China, **4** School of Life Sciences, Tsinghua University, Beijing, China, **5** NHC Key Laboratory of Tropical Disease Control, Hainan Medical University, Haikou, China, **6** Department of Immunology, School of Medicine, the University of Connecticut Health Center, Farmington, Connecticut, United States of America

☯ These authors contributed equally to this work.
* gongcheng@mail.tsinghua.edu.cn

**Data Availability Statement:** All relevant data are within the manuscript and its Supporting Information files.

**Funding:** This work was funded by grants from the National Key Research and Development Plan of

## Abstract

Arboviruses are etiological agents of various severe human diseases that place a tremendous burden on global public health and the economy; compounding this issue is the fact that effective prophylactics and therapeutics are lacking for most arboviruses. Herein, we identified 2 bacterial lipases secreted by a *Chromobacterium* bacterium isolated from *Aedes aegypti* midgut, *Chromobacterium* antiviral effector-1 (*Cb*AE-1) and *Cb*AE-2, with broad-spectrum virucidal activity against mosquito-borne viruses, such as dengue virus (DENV), Zika virus (ZIKV), Japanese encephalitis virus (JEV), yellow fever virus (YFV) and Sindbis virus (SINV). The *Cb*AEs potently blocked viral infection in the extracellular milieu through their lipase activity. Mechanistic studies showed that this lipase activity directly disrupted the viral envelope structure, thus inactivating infectivity. A mutation in the lipase motif of *Cb*AE-1 fully abrogated the virucidal ability. Furthermore, *Cb*AEs also exert lipase-dependent entomopathogenic activity in mosquitoes. The anti-arboviral and entomopathogenic properties of *Cb*AEs render them potential candidates for the development of novel transmission control strategies against vector-borne diseases.

## Author summary

Mosquito-borne viruses are the etiological agents of severe human diseases and annually lead to a great number of deaths. These viruses have spread widely and raised major public health concerns throughout the world. Although effective vaccines have been developed for a few mosquito-borne viruses, such as JEV and yellow fever virus (YFV), vaccines or antiviral therapeutics against most mosquito-borne viruses are currently unavailable. In this study, we identified two virucidal and entomopathogenic effectors with lipase activity, *Cb*AE-1 and *Cb*AE-2, from a mosquito midgut derived bacterium *Chromobacterium sp*. Beijing. Both *Cb*AEs showed potent virucidal activity against a variety of mosquito-borne viruses, including DENV, ZIKV, JEV, YFV, and SINV, as well as other enveloped viruses.

China (2021YFC2300200-4, 2020YFC1200104, 2018YFA0507202, and 2019YFC1200201) (to G.C. and Y.Z.), the Natural Science Foundation of China (81730063, 31825001, 32188101, 82102389 and 81961160737) (to G.C. and Y.Z.), Tsinghua University Spring Breeze Fund (2020Z99CFG017) (to G.C.), the Open Foundation of NHC Key Laboratory of Tropical Disease Control, Hainan Medical University (2021NHCTDCKFKT22013) (to Y.Z.), the Shenzhen San-Ming Project for prevention and research on vector-borne diseases (SZSM201611064) (to G.C.), the Yunnan Chenggong expert workstation (202005AF150034) (to G.C.), G.C. Innovation Team Project of Yunnan Science and Technology Department (202105AE160020) (to G.C.) and the Young Elite Scientists Sponsorship Program (2021QNRC001) (to Y.Z.).The funders had no role in study design, data collection and analysis, decision to publish, or preparation of the manuscript.

**Competing interests:** The authors have declared that no competing interests exist.

Since *Cb*AEs inactivate viruses through their lipase activity by directly disrupting the viral envelope structure, they may provide a novel option for genetically engineering microbiota symbiotic with mosquitoes for arboviral control. Overall, the anti-arboviral and entomopathogenic properties of *Csp_BJ* and *Cb*AEs render them particularly interesting candidates for the development of novel transmission control strategies against vector-borne diseases.

## Introduction

Arboviruses (arthropod-borne viruses), mainly in the Togaviridae, Flaviviridae, Reoviridae, and multiple families in the order Bunyavirales [1], are the etiological agents of severe human diseases, including hemorrhagic fever, biphasic fever, arthritis, encephalitis, and meningitis, and annually lead to a great number of deaths [2–4]. Among these arboviruses, many mosquito-borne viruses have spread widely and raised major public health concerns throughout the world. Dengue virus (DENV), dominantly transmitted by *Aedes* mosquitoes, is currently the most prevalent arbovirus afflicting tropical and subtropical countries worldwide, causing an estimated 390 million infections each year [5]. Zika virus (ZIKV) is another *Aedes*-transmitted arbovirus that has recently re-emerged and poses a threat to global public health [6]. In addition, many neurotropic viruses, such as Japanese encephalitis virus (JEV), West Nile virus (WNV) and Sindbis virus (SINV), transmitted by *Culex* mosquitoes, have also caused high mortality and substantial morbidity in past decades [7–9]. Although effective vaccines have been developed for a few mosquito-borne viruses, such as JEV and yellow fever virus (YFV), vaccines or antiviral therapeutics against most mosquito-borne viruses are currently unavailable [10]. Therefore, the identification of broad-spectrum anti-arboviral agents may offer a solution to control arboviral transmission and prevalence in arthropod vectors. Herein, we identified 2 secreted bacterial proteins from a *Chromobacterium* bacterium that can not only robustly inhibit infections by arboviruses, including DENV and ZIKV, but also manifest entomopathogenic activities in mosquitoes, suggesting their potential as candidates for the development of arboviral control strategies.

## Results

Gut microorganisms in arthropod vectors have been proven to play intricate roles in regulating vector susceptibility to arboviruses in direct or indirect manners [11]. *Chromobacterium spp.* constitute a genus of Gram-negative bacteria that are occasionally pathogenic to humans and animals. A soil bacterium named *Chromobacterium sp.* Panama (*Csp_P*) identified from the midgut of a field-caught *Aedes aegypti* showed strong virucidal activity against DENV [12–14]. In this study, we first identified a *Chromobacterium* from the gut of *A. aegypti* mosquitoes reared in our insectary facility. We thus refer to this bacterium as *Chromobacterium sp.* Beijing (*Csp_BJ*). The absolute abundance of *Chromobacterium* bacterium inside the *A. aegypti* midgut was approximately 200 colonies per midgut, and the relative abundance was 0.22%. We then sequenced the entire bacterial genome to characterize *Csp_BJ* and uploaded the complete genome information into the NCBI database (NCBI Taxonomy ID: 2735795). Based on genomic comparison, *Csp_BJ* shares 99.48% identity with *Chromobacterium haemolyticum CH06-BL*, 99.42% identity with *Chromobacterium rhizoryzae* strain *JP2-74* or 99.52% identity with *Chromobacterium sp.* Panama. To test whether *Csp_BJ* exhibits virucidal activity in mosquitoes, a final concentration of $10^4$–$10^6$ CFU/mL *Csp_BJ* culture (25% v/v) was mixed with human blood (50% v/v) and DENV or ZIKV virus supernatant (25% v/v) for membrane blood

feeding of aseptic *A. aegypti*, and then the infection rate and viral titre in mosquitoes were determined by qPCR (Fig 1A and 1D). Oral supplementation of this bacterium in *A. aegypti* not only largely impaired the infection rates of DENV (Fig 1B) and ZIKV (Fig 1E) in mosquitoes but also reduced the viral titre in mosquitoes in a dose-dependent manner (Fig 1C and 1F). It is worth noting that entomopathogenic activity of *Csp_BJ* was also observed in this experiment. To further confirm its entomopathogenic activity, a final concentration of $10^4$–$10^8$ CFU/mL *Csp_BJ* culture (50% v/v) was mixed with either 1% sugar solution or human blood (50% v/v) for membrane feeding of aseptic *A. aegypti* (Fig 1G). Exposure to *Csp_BJ* at concentrations higher than $10^5$ CFU/mL resulted in high mortality rates in *A. aegypti* females (Fig 1H and 1I).

We next aimed to understand the mechanism underlying the antiviral activity of *Csp_BJ* in mosquitoes. Bacteria usually exploit many effectors, such as cellular components, metabolites or secreted proteins, to regulate their host immune or physiological status for effective colonization. We therefore identified the bacterial effector(s) that modulate DENV and ZIKV infection by separating the *Csp_BJ* culturing suspension into different fractions. In this experiment, we cultured *Csp_BJ* for 24 hr at 30°C. The cell-free culture supernatant was collected by centrifugation and ensuing filtration through a 0.22 μm filter unit, whereas the cell lysates were generated by sonication. Either the bacterial cell lysate or the culture supernatant (25% v/v) was mixed with human blood (50% v/v) and DENV or ZIKV virus supernatant (25% v/v) for membrane blood feeding of *A. aegypti* (Fig 2A). Ingestion of the culture supernatant, but not the bacterial lysates, resulted in significant suppression of DENV (Fig 2C) and ZIKV (Fig 2D) infection rates in *A. aegypti*, indicating that an extracellular effector(s) secreted by *Csp_BJ* was responsible for viral inhibition. Next, we investigated whether the effector(s) were secreted proteins, small peptides, lipids, polysaccharides or other metabolites. Therefore, the culture supernatant was separated using a 3 kDa cutoff filter [15]. Either the upper retentate (proteins and large peptides) or the lower liquid filtrate (small molecule compounds and short peptides) (25% v/v) was mixed with human blood (50% v/v) and the viruses (25% v/v) for membrane blood feeding (Fig 2B). Intriguingly, ingestion of the retentate rather than the filtrate inhibited the infection rates of DENV (Fig 2E) and ZIKV (Fig 2F), suggesting that the effector(s) might be a protein(s) secreted by *Csp_BJ*. Furthermore, the same components were mixed with 50 plaque-forming units (PFU) of DENV or ZIKV and incubated for 1 hr, and then the infectious viral particles were determined by a plaque forming assay (Fig 2A and 2B). Consistently, incubation of both the culture supernatant and the upper retentate inhibited DENV (Fig 2G and 2I) and ZIKV (Fig 2H and 2J) infectivity in Vero cells, indicating that the effector(s) acts against the viruses in a direct manner.

To further identify the effector(s), the protein components in the upper retentate were separated by SDS–PAGE and then analyzed by mass spectrometry (Fig 3A). Highly abundant proteins with secretable properties were selected, expressed and purified in an *Escherichia coli* expression system (Fig 3B). Of all the proteins tested, the bacterial protein encoded by gene3771 (Accession: MT473992) significantly impaired DENV and ZIKV infection in Vero cells (Fig 3C and 3D). We named this protein *Chromobacterium* antiviral effector-1 (*Cb*AE-1). Intriguingly, a *Cb*AE-1 homolog with 55.61% amino acid identity named *Cb*AE-2 (Accession: MT473995) was further identified from *Csp_BJ* based on sequence comparison (S1 Fig). Both effectors encoded a lipase domain with a typical GDSL motif [16]. To further confirm the virucidal activity, both proteins were expressed and purified in *E. coli* (Fig 3E). Serial concentrations of recombinant proteins were mixed with 50 PFU of DENV or ZIKV for the plaque assay in Vero cells. The half-inhibitory concentration ($IC_{50}$) of *Cb*AE-1 was 0.0016 μg/mL for DENV and 0.0026 μg/mL for ZIKV (Fig 3F and 3G). However, the $IC_{50}$ of *Cb*AE-2 for both flaviviruses was 370–2,400 times higher than that of *Cb*AE-1, indicating much more robust

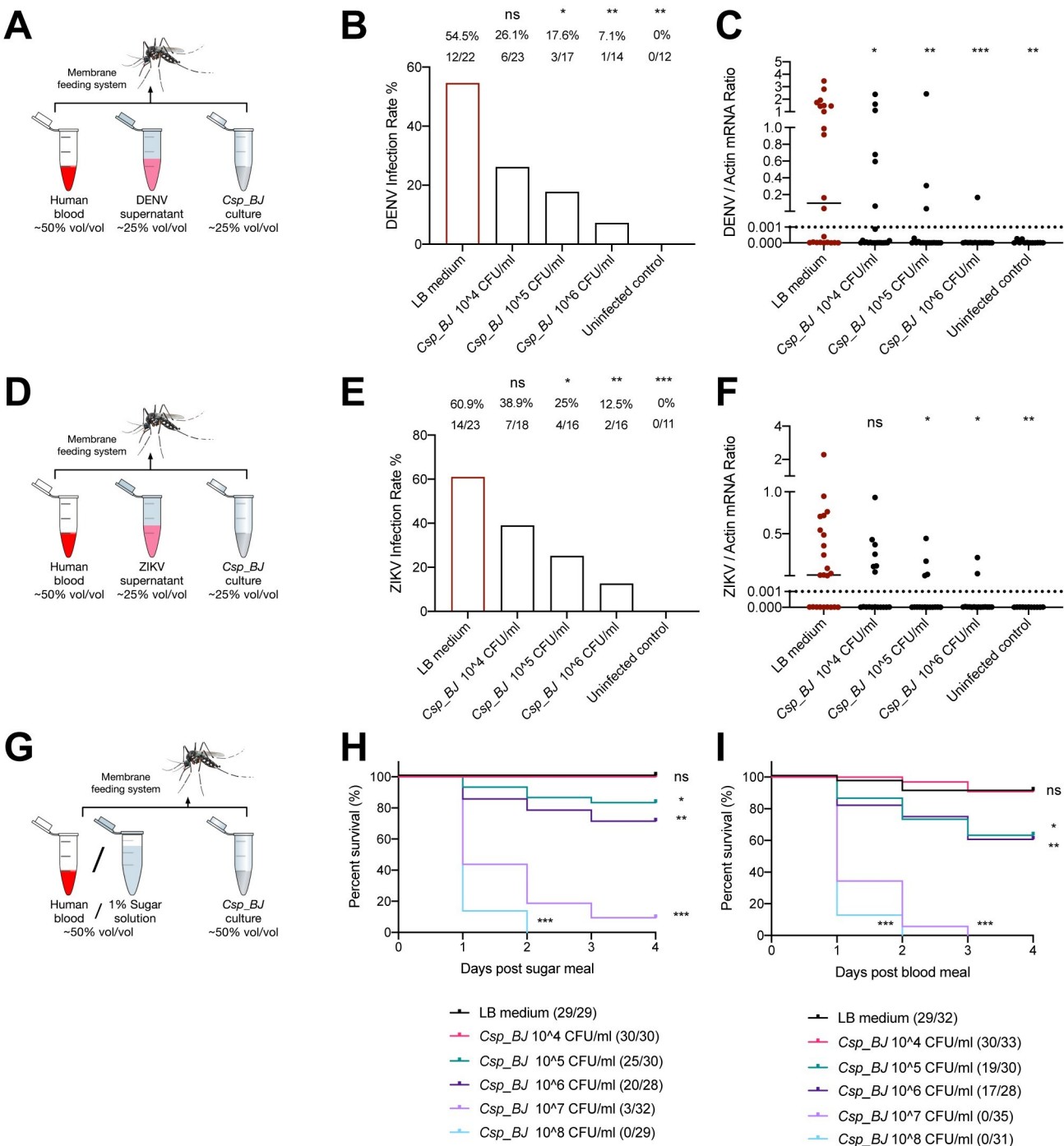

**Fig 1. *Csp_BJ* shows virucidal activity against DENV and ZIKV as well as entomopathogenic activity in *A. aegypti*.** (A-F) Oral supplementation with *Csp_BJ* inhibits DENV (A-C) and ZIKV (D-F) infection of *A. aegypti*. (A, D) Schematic representation of the study design. A mixture containing human blood (50% v/v), *Csp_BJ* bacterial suspension (25% v/v), and supernatant from DENV- (A) or ZIKV- (D) infected Vero cells (25% v/v) was used to feed antibiotic-treated *A. aegypti* Rockefeller strain via a membrane blood feeding system. Mixtures containing human blood (50% v/v), fresh LB broth (25% v/v), and supernatant from DENV- or ZIKV-infected Vero cells (25% v/v) served as negative controls. A mixture containing human blood (50% v/v), *Csp_BJ* bacterial suspension (25% v/v), and fresh VP-SFM medium (25% v/v) served as an uninfected control. Mosquito infectivity was determined by RT–qPCR at 8 days post blood meal. The final DENV or ZIKV titre was $1 \times 10^5$ PFU/mL for oral infection. (B, E) Mosquito infection rate after oral supplementation of *Csp_BJ* culture with DENV (B) or ZIKV (E). (C, F) Mosquito viral load after oral supplementation of *Csp_BJ* culture with DENV (C) or ZIKV (F). (G-I) Oral supplementation with *Csp_BJ* caused a high mortality rate in *A. aegypti*. (G) Schematic representation of the study design. A mixture containing 1% sugar solution (H) or human blood (I) (50% v/v) and *Csp_BJ* bacterial suspension (50% v/v) was used to feed antibiotic-treated *A. aegypti* via a membrane blood feeding system. Mixtures containing 1% sugar solution or human blood (50% v/v) and fresh LB broth (50% v/v) served as

negative controls. Mosquito mortality was observed over 4 days. (B, C, E, F, H, I) The number of infected mosquitoes relative to total mosquitoes is shown at the top of each column (B, E). Differences in the infectivity ratio were compared using Fisher's exact test (B, E). A nonparametric Mann–Whitney test was used for the statistical analysis (C, F). The survival rates of mosquitoes were plotted using a Kaplan–Meier curve and were statistically analyzed using the log-rank (Mantel-Cox) test (H, I). $P$ values were adjusted using the Benjamini–Hochberg procedure (B, E, H, I) or Dunnett's test (C, F) to account for multiple comparisons. The $P$ value represents a comparison between the control group and the other groups. *$P < 0.05$, **$P < 0.01$, ***$P < 0.001$, ****$P < 0.0001$, ns, not significant. The limit of detection is illustrated by dotted lines (C, F). Experiments consisted of at least three biological replicates, with similar results observed among the trials.

virucidal activity of *Cb*AE-1 against flaviviruses (Fig 3F and 3G). Thus, we identified 2 bacterial effectors with high virucidal activity from a *Csp_BJ* bacterium.

We next investigated the mechanisms by which these bacterial effectors inhibit viral infection. According to sequence analysis, *Cb*AEs contain a conserved lipase domain. Lipases are a group of enzymes that catalyze the hydrolysis of the ester bond(s) of glycerides into fatty acids and glycerol [16]. We therefore assessed whether *Cb*AEs exhibit lipase activity. In a plate degradation assay, both *Cb*AEs directly digested egg yolk lipids and formed lytic halos whose diameters correlated with lipase activity (Fig 4A). We then repeated the plaque reduction neutralization assays of *Cb*AEs with phospholipase A2 (PLA$_2$) from *A. mellifera* honeybee venom with known antiviral activity and PLA$_2$ from bovine pancreas. *A. mellifera* PLA$_2$ showed similar antiviral activities against the viruses as previously reported [17] and the IC$_{50}$ values of *Cb*AEs were consistent with our previous results (S2 Fig). The bovine pancreas PLA$_2$ did not show any antiviral activity (S2 Fig), indicating the specific virucidal activities of *Cb*AEs and *A. mellifera* PLA$_2$. The sequence GDSL is the core motif of lipase activity [16]. Consistently, a S187G mutation in this motif of *Cb*AE-1 fully disrupted its lipase activity (Fig 4A), validating *Cb*AEs as secreted lipases of *Csp_BJ*. Given their lipase activity, we hypothesized that *Cb*AEs might use their enzymatic activity to degrade the viral lipid envelope, which may result in exposure to viral RNA [18]. To address this hypothesis, serial concentrations of *Cb*AEs were incubated with $1 \times 10^4$ PFU of DENV or ZIKV for 1 hr at 37˚C. Each mixture was then treated with RNase A to evaluate the degradation of exposed viral genomic RNA. Compared to mock treatments in which the viruses were incubated with GFP, a significant reduction in viral RNA was recorded by RT–qPCR when the viruses were treated with *Cb*AEs (Fig 4B and 4C), indicating that the lipase activity of *Cb*AEs directly disrupted the virion structure, thus resulting in viral genome release. Consistent with these results, the S187G mutant of *Cb*AE-1 that had no lipase activity completely failed to suppress both DENV and ZIKV infection in Vero cells (Fig 4D and 4E), further indicating that the virucidal activity of *Cb*AEs is lipase-dependent. To validate that *Cb*AEs disrupt viral lipidic membranes, we incubated *Cb*AE-1 with purified ZIKV virions and processed the samples for transmission electron microscopy (Fig 4F). ZIKV particles typically have diameters of 50–60 nm [19], and consistent with that, we observed intact viral particles in our control sample treated with 10 μg/mL BSA (Fig 4F). However, upon treatment with 10 μg/mL *Cb*AE-1 in the same BSA solution buffer, the integrity of ZIKV particles was fully disrupted (Fig 4F). This is in agreement with our previous finding that treatment with *Cb*AEs resulted in viral genome release. Since *Cb*AEs blocked viral infection in the extracellular milieu, we further assessed whether treatment with *Cb*AEs prior to viral inoculation could effectively block viral infection. We pretreated Vero cells with 10 μg/mL purified *Cb*AEs. Subsequently, 0.1 MOI of DENV or ZIKV was used to challenge the *Cb*AE-treated cells. Preincubation with *Cb*AE-1 fully blocked the infectivity of both flaviviruses, whereas treatment with *Cb*AE-2 exhibited 60%-80% inhibition (Fig 4G and 4H). Next, we generated a truncated form of *Cb*AE-1 which only consists of its GDSL lipase domain (181AA-339AA), which is around 29kDa. We then tested the antiviral activity of *Cb*AE-1-truncated against DENV and ZIKV. The lipase domain alone did show antiviral activity against both DENV and ZIKV. However,

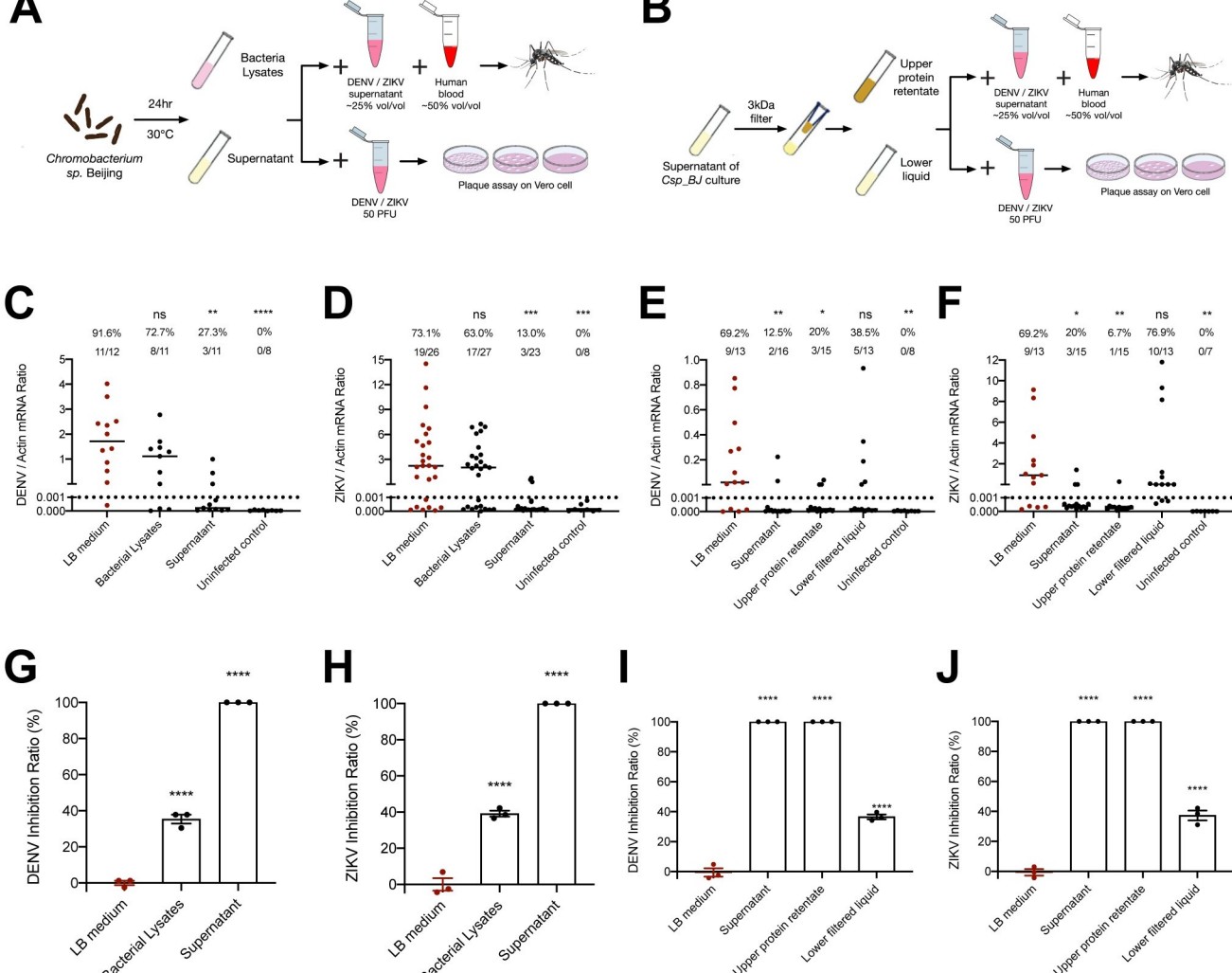

**Fig 2. Secreted effector(s) from *Csp_BJ* inhibit DENV and ZIKV infection in both *A. aegypti* and Vero cells.** (A, B) Schematic representation of the study design. (A) A *Csp_BJ* suspension was separated into bacterial cells and culture supernatant by centrifugation. Either the bacterial cell lysate or the culture supernatant (25% v/v) was mixed with human blood (50% v/v) and DENV or ZIKV virus supernatant (25% v/v) for membrane blood feeding of antibiotic-treated *A. aegypti*. Either the bacterial lysates or the culture supernatant (50% v/v) mixed with 50 PFU of DENV or ZIKV in VP-SFM medium (50% v/v) were incubated for 1 hr before being used for infection of Vero cell monolayers. Positive and negative controls are described in details in the Materials and Methods section. (B) Either the upper retentate or the lower liquid filtrate (25% v/v) was mixed with human blood (50% v/v) and the viruses (25% v/v) for membrane blood feeding of antibiotic-treated *A. aegypti*. Either the retentate or the filtrate (50% v/v) was mixed with 50 PFU of DENV or ZIKV in VP-SFM medium (50% v/v) and incubated for 1 hr before inoculation into Vero cell monolayers. Positive and negative controls are described in details in the Materials and Methods section. (A, B) Mosquito infectivity was determined by RT–qPCR at 8 days post blood meal. The final DENV or ZIKV titre was $1 \times 10^5$ PFU/mL for oral infection. (C, D, G, H) Secreted factor(s) from *Csp_BJ* inhibited DENV and ZIKV infection in mosquitoes or Vero cells. (C, D) Mosquito infection rate after oral supplementation of cell lysates or culture supernatant of *Csp_BJ* along with DENV (C) or ZIKV (D). (G, H) Inhibition rate in the presence of cell lysates or culture supernatant of *Csp_BJ* following infection by DENV (G) or ZIKV (H) in Vero cells, as determined by plaque formation assay. (E, F, I, J) Proteins in the supernatant inhibited DENV and ZIKV infection in mosquitoes or Vero cells. (E, F) Mosquito infection rate after oral supplementation of different components of *Csp_BJ* culture along with DENV (E) or ZIKV (F). (I, J) The inhibition rate in Vero cells was determined by plaque formation assay. (C-J) The number of infected mosquitoes relative to total mosquitoes is shown at the top of each column. Differences in the infectivity ratio were compared using Fisher's exact test (C-F). Significance was determined by unpaired t tests (G-J). Data are presented as the mean ± SEM (G-J). *P* values were adjusted using the Benjamini–Hochberg procedure (C-F) or Dunnett's test (G-J) to account for multiple comparisons. The *P* value represents a comparison between the mock group and the other groups. $^*P < 0.05$, $^{**}P < 0.01$, $^{***}P < 0.001$, $^{****}P < 0.0001$, ns, not significant. The limit of detection is illustrated by dotted lines (C-F). Experiments consisted of at least three biological replicates, with similar results observed among the trials.

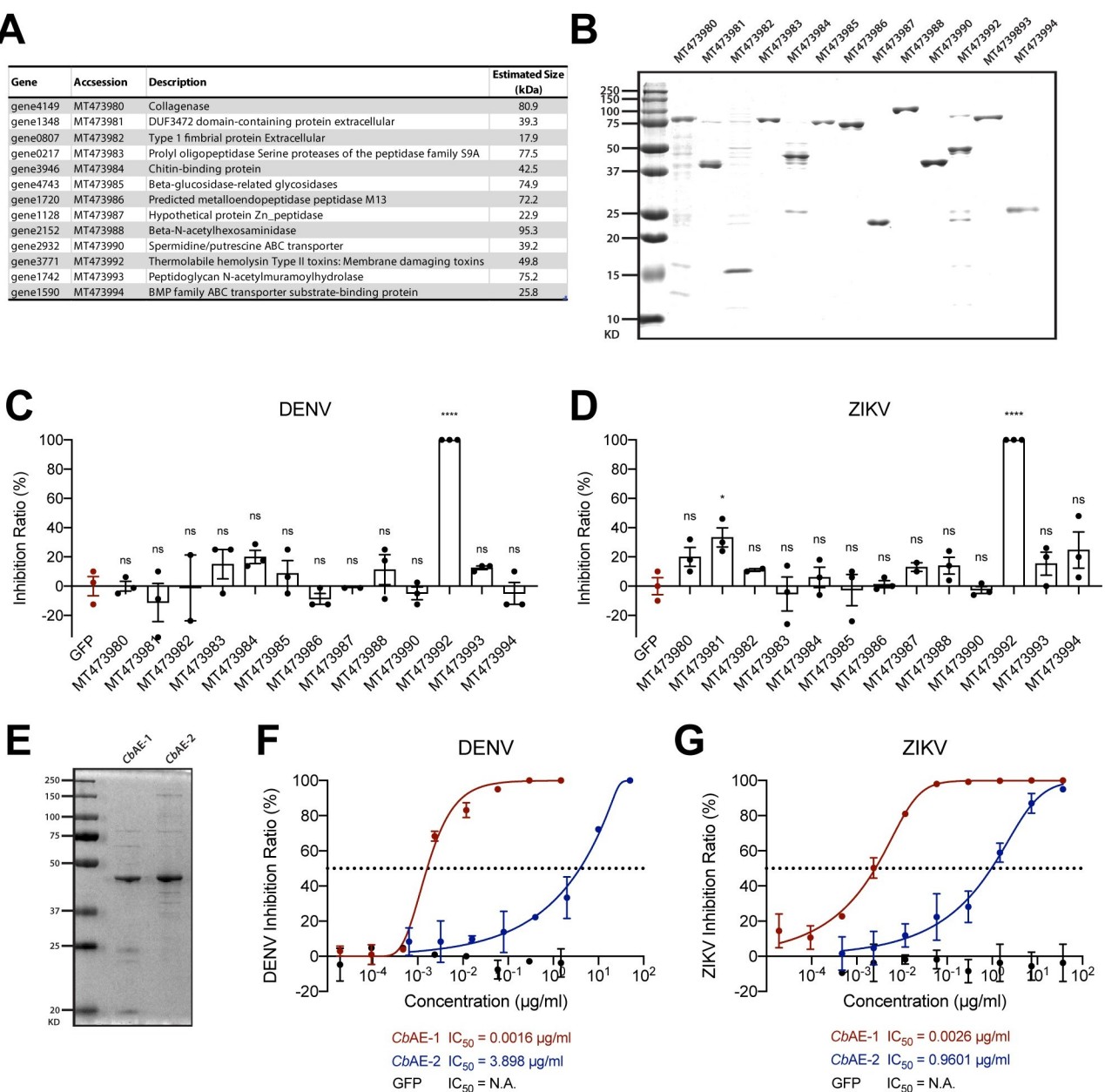

**Fig 3. Identification of *Csp_BJ*-secreted antiviral effector(s) against DENV or ZIKV.** (A) The protein description was obtained from the UniProt and NCBI databases. (B) The identified *Csp_BJ* proteins were expressed and purified from *E. coli* cells. (C, D) A total of 1 μg of purified recombinant protein was mixed with 50 PFU of DENV (C) or ZIKV (D) in VP-SFM medium and incubated for 1 hr before infecting Vero cell monolayers. (E-G) *Csp_BJ* inhibited DENV and ZIKV infections via its secreted proteins *Cb*AE-1 and *Cb*AE-2. (E) The identified *Csp_BJ* secreted proteins *Cb*AE-1 and *Cb*AE-2 were expressed and purified from *E. coli* cells. (F, G) Inhibition curves of *Cb*AE-1 and *Cb*AE-2 against DENV (F) and ZIKV (G). Serial concentrations of *Cb*AE-1 or *Cb*AE-2 were mixed with 50 PFU of DENV or ZIKV in VP-SFM medium to perform standard plaque reduction neutralization tests (PRNTs). (C, D) Significance was determined using unpaired t tests. Data are presented as the mean ± SEM. *P* values were adjusted using Dunnett's test to account for multiple comparisons. The *P* value represents a comparison between the control group and the other groups. $^*P < 0.05$, $^{****}P < 0.0001$, ns, not significant. Experiments consisted of at least three biological replicates with similar results.

the half-inhibitory concentration ($IC_{50}$) of *Cb*AE-1-truncated was 20–80 times higher than that of *Cb*AE-1, indicating impaired virucidal activity of *Cb*AE-1-truncated (S3 Fig).

Since the *Cb*AEs showed direct enzymatic action on the viral lipid bilayer, we assessed their virucidal activity against other mosquito-borne viruses. The culture supernatant of JEV, YFV

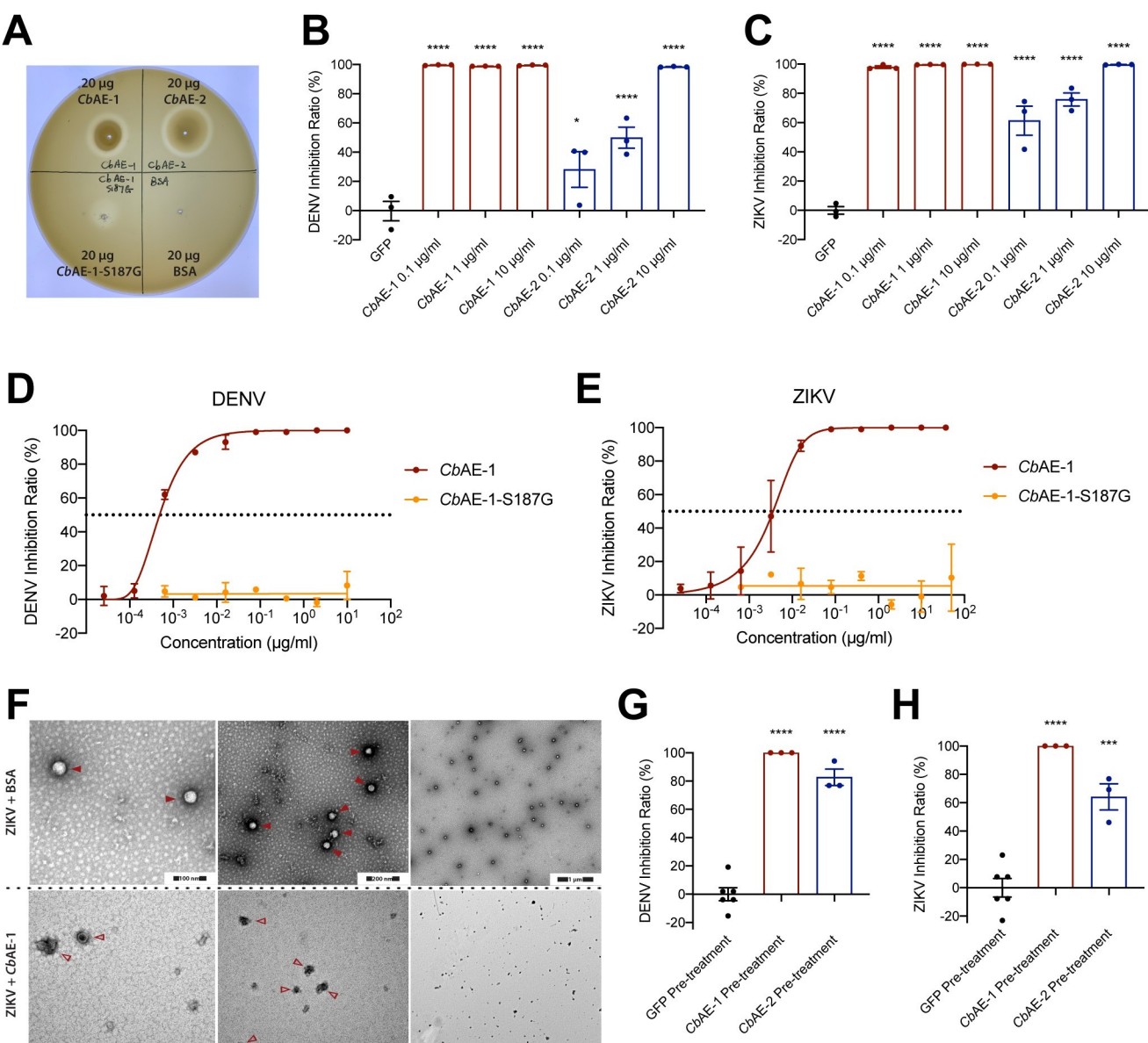

**Fig 4. The virucidal activity of *Cb*AEs is mediated by enzymatic degradation of the viral lipidic envelope.** (A) Lipase enzymatic activity of *Cb*AE-1, *Cb*AE-2 and *Cb*AE-1-S187G measured via an egg yolk agar plate assay. (B, C) Analysis of the exposure of DENV (B) or ZIKV (C) genomic RNA. DENV or ZIKV was first treated with serial concentrations of *Cb*AE-1, *Cb*AE-2 and then with RNase-A. Viral RNA degradation was evaluated by RT–qPCR. (D, E) The S187G mutant of *Cb*AE-1 fully lost its ability to suppress DENV (D) and ZIKV (E) infection: inhibition curves of *Cb*AE-1 and *Cb*AE-1-S187G against DENV (D) or ZIKV (E). Serial concentrations of *Cb*AE-1 or *Cb*AE-1-S187G were mixed with 50 PFU of DENV or ZIKV in VP-SFM medium to perform standard plaque reduction neutralization tests (PRNTs). (F) Representative negative stained transmission electron microscopy images of ZIKV particles treated with 10 μg/mL BSA (arrowhead) and those treated with 10 μg/mL *Cb*AE-1 (empty arrowhead); high magnification: 240,000×, medium magnification: 120,000×, low magnification: 30,000×. (G, H) Rate of DENV (G) or ZIKV (H) replication inhibition following exposure to *Cb*AEs before viral infection of Vero cell monolayers. The viral genome was quantified by RT–qPCR. (B, C, G, H) Significance was determined using unpaired t tests. Data are presented as the mean ± SEM. *P* values were adjusted using Dunnett's test to account for multiple comparisons. The *P* value represents a comparison between the control group and the other groups. $^{**}P < 0.01$, $^{***}P < 0.001$, $^{****}P < 0.0001$. Experiments consisted of at least three biological replicates with similar results.

or SINV was incubated with a serial concentration of *Cb*AEs prior to plaque assays in Vero cells. Both *Cb*AEs were able to directly inhibit JEV, YFV and SINV infections, with $IC_{50}$ values ranging from 0.008 μg/mL to 0.023 μg/mL for *Cb*AE-1 and from 0.139 μg/mL to 10.70 μg/mL

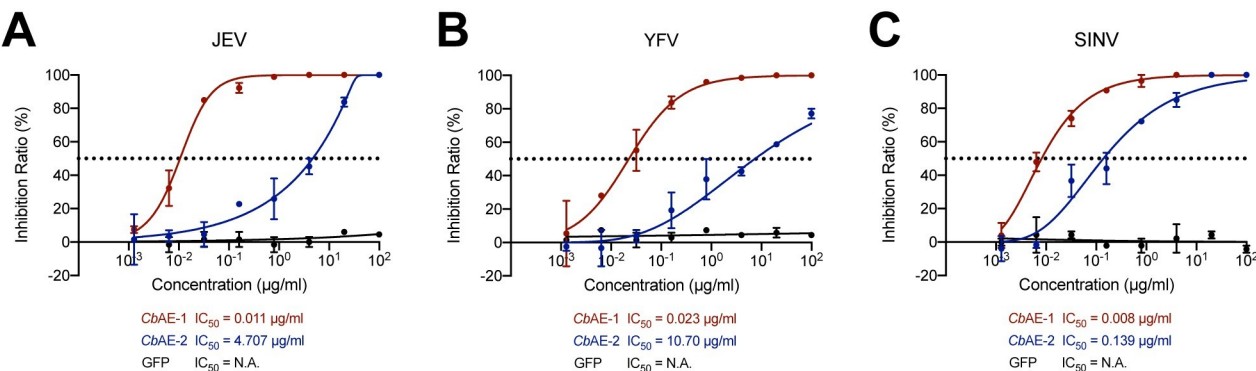

**Fig 5. The virucidal activity of *Cb*AEs against JEV, YFV and SINV.** (A-C) Inhibition curves of *Cb*AE-1 and *Cb*AE-2 against JEV (A), YFV (B), and SINV (C). Standard plaque reduction neutralization tests (PRNTs) were performed.

for *Cb*AE-2 (Fig 5). We then tested whether *Cb*AEs have antiviral activities against other enveloped viruses except for these mosquito-borne viruses. The antiviral activities of *Cb*AE-1 and *Cb*AE-2 against SARS-CoV-2 pseudovirus and HSV-1 were assessed on Vero cells respectively. Both *Cb*AEs showed varying degrees of virucidal activity against them (S4 Fig). Since *Cb*AEs have shown antiviral activities against enveloped viruses or pseudovirus from the families Flaviviridae, Togaviridae, Coronaviridae, and Herpesviridae, it is rational to hypothesize that the virucidal activity of *Cb*AEs is mediated through the same mechanism of lipase activity. Considering *Cb*AEs disrupt the lipid associate membranes, we tested the cytotoxicity of *Cb*AE-1 and *Cb*AE-2 on Vero cells and C6/36 cells, respectively. The half-cytotoxicity concentrations ($CC_{50}$) of *Cb*AE-1 and *Cb*AE-2 in both cell lines are indicated in the figures (S5 Fig). Only *Cb*AE-1 showed toxicity at higher concentrations in Vero cells. At concentrations used in experiments generating $IC_{50}$ curves, neither *Cb*AE-1 nor *Cb*AE-2 showed any cytotoxicity against Vero cells or C6/36 cells (S5 Fig).

To verify whether *Cb*AEs mediate virucidal and entomopathogenic activities in mosquitoes through their lipase activity, either *Cb*AE-1 or its S187G mutant was orally supplemented with *A. aegypti* (Fig 6A, 6D and 6G). Consistent with previous *in vitro* studies, ingestion of 10 µg/mL *Cb*AE-1, but not its S187G mutant, resulted in a significant reduction in *A. aegypti* permissiveness with respect to DENV (Fig 6B and 6C) and ZIKV (Fig 6E and 6F). In addition, while oral supplementation of 10 µg/mL *Cb*AE-1 with either 1% sugar solution or human blood resulted in high mortality rates of *A. aegypti*, the S187G mutation completely abolished its entomopathogenic activity (Fig 6H and 6I), suggesting that the entomopathogenic property of *Cb*AEs is lipase-dependent. These experiments were then repeated with mosquitoes untreated by antibiotics. The virucidal and entomopathogenic activities of *Cb*AE-1 and *Cb*AE-1-S187G were consistent with the results observed in aseptic mosquitoes (S6 Fig).

## Discussion

In this study, we identified two virucidal and entomopathogenic effectors with lipase activity, *Cb*AE-1 and *Cb*AE-2, from *Chromobacterium sp. Csp_BJ*. Both *Cb*AEs showed potent virucidal activity against a variety of mosquito-borne viruses, including DENV, ZIKV, JEV, YFV, and SINV. Indeed, accumulating evidence indicates that certain lipases present potent antiviral activity. Either lipoprotein lipase or hepatic triglyceride lipase impaired hepatitis C virus (HCV) infection in human Huh7.5 cells by degrading virus-associated lipoproteins [20]. A secreted PLA$_2$ isolated from *Naja mossambica* snake venom showed potent virucidal activity

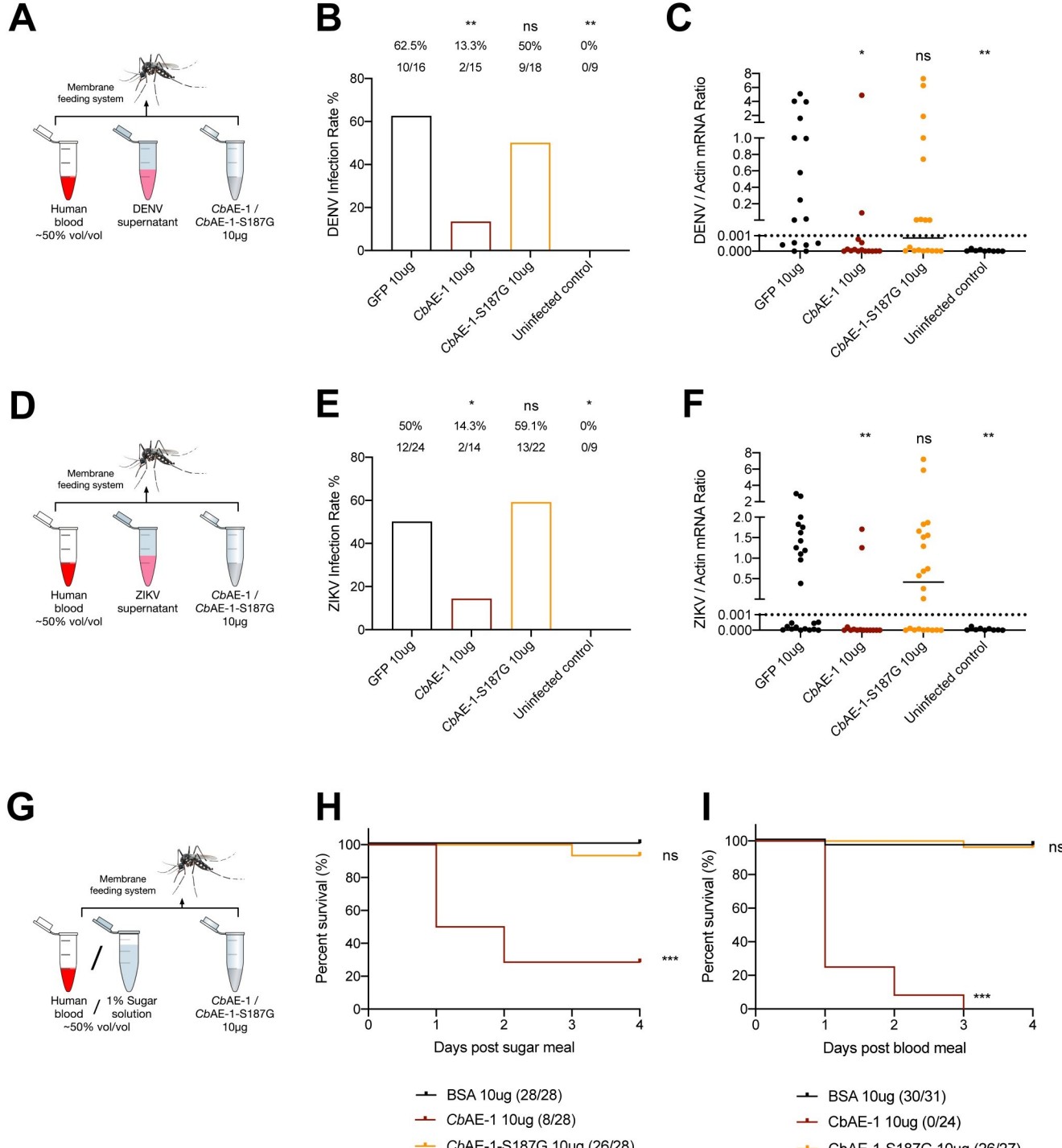

**Fig 6. Both the virucidal and entomopathogenic properties of *Cb*AE-1 in mosquitoes are dependent on its lipase activity.** (A-F) The virucidal activity of *Cb*AE-1 against DENV (A-C) and ZIKV (D-F) infection in *A. aegypti* is dependent on its lipase activity. (A, D) Schematic representation of the study design. A mixture containing human blood (50% v/v), 10 μg *Cb*AE-1 or *Cb*AE-1-S187G, and supernatant from DENV- (A) or ZIKV- (D) infected Vero cells was used to feed antibiotic-treated *A. aegypti* via a membrane blood feeding system. Mosquito infectivity was determined by RT–qPCR at 8 days post blood meal. The final DENV or ZIKV titre was $1 \times 10^5$ PFU/mL for oral infection. (B, E) Mosquito infection rate after oral supplementation of *Cb*AE-1 or *Cb*AE-1-S187G with DENV (B) or ZIKV (E). (C, F) Mosquito viral load after oral supplementation of *Cb*AE-1 or *Cb*AE-1-S187G with DENV (C) or ZIKV (F). (G-I) The entomopathogenic activity of *Cb*AE-1 in *A. aegypti* is dependent on its lipase activity. (G) Schematic representation of the study design. A mixture containing 1% sugar solution (H) or human blood (I) (50% v/v) and 10 μg *Cb*AE-1 or *Cb*AE-1-S187G (50% v/v) was used to feed antibiotic-treated *A. aegypti* via a membrane blood feeding system. Mosquito mortality was observed over 4 days. (B, C, E, F, H, I) The number of infected mosquitoes relative to

total mosquitoes is shown at the top of each column (B, E). Differences in the infectivity ratio were compared using Fisher's exact test (B, E). A nonparametric Mann–Whitney test was used for the statistical analysis (C, F). The survival rates of mosquitoes were plotted using a Kaplan–Meier curve and were statistically analyzed using the log-rank (Mantel-Cox) test (H, I). $P$ values were adjusted using the Benjamini–Hochberg procedure (B, E, H, I) or Dunnett's test (C, F) to account for multiple comparisons. The $P$ value represents a comparison between the control group and the other groups. [*]$P < 0.05$, [***]$P < 0.001$, [****]$P < 0.0001$, ns, not significant. The limit of detection is illustrated by dotted lines (C, F). Experiments consisted of at least three biological replicates with similar results.

against HCV, DENV, and JEV, while the protein did not exhibit significant antiviral activity against SINV, IAV, Middle East respiratory syndrome coronavirus (MERS-CoV) or HSV-1 [17]. A secreted human $PLA_2$ has been shown to neutralize HIV-1 by degrading the viral membrane [21] or by blocking viral entry into host cells, rather than through a lipase-mediated virucidal effect [22], suggesting diverse virucidal mechanisms of $PLA_2$. It is worth noting that while *Cb*AE-1 and *Cb*AE-2 share 52% identity, their antiviral activity differs significantly. According to a previous study on three isoforms of phospholipase A2 (CB1, CB2 and $PLA_2$-IC) from *Crotalus durissus terrificus* snake venom, while CB1 and CB2 share 91% and 97% identity with $PLA_2$-IC, respectively, the enzymatic activities of these three isoforms differ significantly [23]. Another study of $PLA_2$ from snake venom showed that B*l*K-$PLA_2$ and B*l*D-$PLA_2$ had 58% similarity in amino acid sequence but also differed in antiviral activity against dengue virus [24]. It is possible that certain regions in these lipases play an important role in regulating their specificity or enzymatic activity. The difference in specificity or enzymatic activity resulted in variation of antiviral activities. The $PLA_2$ superfamily is currently classified into six types, in which more than one-third of the members belong to $sPLA_2$ (secreted $PLA_2$), which is further divided into 10 groups and 18 subgroups [25]. These $sPLA_2$s show functional variations and are involved in a wide range of biological functions and disease occurrence through lipid metabolism and signalling [25]. The $sPLA_2$s from bacteria are classified into group XIV, while the $sPLA_2$s from honeybee venom and bovine pancreas belong to group III and subgroup IB, respectively. It is possible that $sPLA_2$s from different groups or subgroups differ in their ability to degrade the viral envelope. However, the exact mechanism underlying this difference still needs further investigation, and we aim to identify factors affecting their specificity and enzymatic activity in future studies.

*Cb*AEs may inactivate viruses through their lipase activity, which also damages cellular membranes; nonetheless, their specificity and affinity for the viral envelope could be significantly improved by rational design and engineering, further reducing their cytotoxicity and increasing virucidal efficacy. On the other hand, since *Cb*AEs directly regulate viral infection and survival rate in mosquitoes, they may provide a novel option for genetically engineering microbiota symbiotic with mosquitoes for arboviral control, which is called paratransgenesis. This strategy has already been tested in a gut endosymbiont of *Rhodnius prolixus* to control the transmission of the Chagas disease-causing parasite *Trypanosoma cruzi* [26] and has been applied to inhibit malaria by engineering common bacteria symbiotic with mosquitoes to express antiplasmodial effectors [27,28]. Therefore, the anti-arboviral and entomopathogenic properties of *Csp_BJ* and *Cb*AEs render them particularly interesting candidates for the development of novel transmission control strategies against vector-borne diseases.

## Materials and methods

### Ethics statement

Human blood was collected from healthy donors who provided written informed consent. The collection of human blood samples and their use for mosquito feeding was approved by the local ethics committee of Tsinghua University.

## Mosquitoes, cells, viruses and bacteria

*Aedes aegypti* (the Rockefeller strain) was maintained on a sugar solution in a low-temperature, illuminated incubator (Model 818, Thermo Electron Corporation) at 28˚C and 80% humidity, according to standard rearing procedures [29]. Vero cells were maintained in Dulbecco's modified Eagle's medium (11965–092, Gibco) supplemented with 10% heat-inactivated fetal bovine serum (16000–044, Gibco) and 1% antibiotic-antimycotic (15240–062, Invitrogen) in a humidified 5% (V/V) $CO_2$ incubator at 37˚C. The Vero and C6/36 cell line were purchased from ATCC (CCL-81 and CRL-1660). DENV-2 (New Guinea C strain), ZIKV (PRVABC59 strain), JEV (SA14 strain), YFV (17D strain), SINV (YN87448) and HSV-1 were grown in Vero cells with VP-SFM medium (11681–020, Gibco). Viruses were titrated by a standard plaque formation assay conducted on Vero cells [30]. *Chromobacterium sp*. Beijing culture suspension was grown in LB broth at 30˚C for 24 hr at 250 rpm.

## Isolation and characterization of mosquito midgut bacteria

Mosquitoes were anesthetized in a 4˚C refrigerator and then surfaced-sterilized by dipping and shaking them in 75% ethanol for 2 min and rinsing them with 1× PBS twice for 1 min each. Midguts were then dissected from each individual mosquito over a sterile glass slide containing a drop of 1× PBS, transferred to a microcentrifuge tube containing 200 μl of sterile PBS and macerated for 30 sec. Three 10-fold serial dilutions were then plated on LB agar and kept at room temperature for 48 hr. The bacterial colonies were identified via 16S rRNA gene sequencing and comparisons [31,32].

## Whole genome sequencing of *Csp_BJ*

The genomic DNA library of *Csp_BJ* was constructed by using a QIAseq FX DNA Library Kit (Qiagen) according to the manufacturer's instructions, and then by paired-end sequencing using an Illumina NextSeq 500 platform with a 300-cycle NextSeq 500 reagent kit v2 (Illumina). The metagenomic samples were sequenced by single-end sequencing by using a 150-cycle NextSeq 500 Reagent Kit v2 (Illumina). The complete genome sequence of the strain was determined by using a PacBio Sequel (Pacific BioSciences) sequencer with a Sequel SMRT Cell 1M v2 (four/tray) and a Sequel sequencing kit v2.1 (Pacific BioSciences) for long-read sequencing (insert size, ≈10 kb). High quality genomic DNA was used to prepare a SMRTbell library by using a SMRTbell Template Prep Kit 2.0 (Pacific Biosciences). The draft genome contigs were assembled by using A5-Miseq software with Illumina short reads [33]. The circular genome sequence was constructed by using Canu version 1.4 [34], Minimap version 0.2-r124 [35], racon version 1.1.0 [36], and Circlator version 1.5.3 [37] with long read data. Error correction of the circular sequence was performed by using Pilon version 1.18 with short reads [38]. Annotation was performed in DFAST version 1.0.8 [39] and NCBI-BLASTP/ BLASTX against deposited *Chromobacterium* complete genome sequences.

## Preparation of *Csp_BJ* culture suspension, culture supernatant, cell lysate, upper retentate and lower liquid filtrate

The concentration of the *Csp_BJ* culture suspension at OD600 = 1.0 was first determined by a colony-forming assay. Briefly, *Csp_BJ* was grown overnight in liquid LB at 30˚C and diluted to OD600 = 1.0 using fresh LB. Then, 10-fold serial dilutions of *Csp_BJ* culture suspension at OD600 = 1.0 were plated on LB agar and kept at 30˚C for 48 hr. The concentration of the *Csp_BJ* culture suspension at OD600 = 1.0 was calculated to be approximately $10^8$ CFU/ml. For mosquito feeding experiments, *Csp_BJ* was grown overnight in liquid LB at 30˚C. The

overnight culture was diluted using fresh LB to OD600 = 1.0, which equals a concentration of approximately $10^8$ CFU/ml, and was further diluted to $10^4$–$10^7$ CFU/ml using fresh LB.

To prepare different components of the *Csp_BJ* culture, *Csp_BJ* was cultured for 24 hr at 30˚C in liquid LB. The *Csp_BJ* suspension was then separated into bacterial cells and culture supernatant by centrifugation at $4000 \times g$ and 4˚C for 10 minutes (Avanti J-26XP, Beckman). The cell-free culture supernatant was then filtered through a 0.22 μm filter unit (SLGP033NS) before use in mosquito feeding experiments or plaque assays. The pelleted bacteria were washed twice in 1× PBS and resuspended in cold 1× PBS, and cell lysates were generated by probe sonication. Lysates were centrifuged at $8000 \times g$ and 4˚C for 20 minutes (Avanti J-26XP, Beckman) to remove insoluble cell debris and were then filtered through a 0.22 μm filter unit before use. The filtered cell-free culture supernatant was further separated using a 3 kDa cutoff filter (ACK5003PA, Millipore) by centrifugation at 4˚C (5810R, A-4-62, Eppendorf). The upper retentate (proteins and large peptides) or the lower liquid filtrate (small molecule compounds and short peptides) were transferred to new tubes for further experiments.

## Generation of antibiotic-treated mosquitoes

*Aedes aegypti* (Rockefeller strain) were maintained on a 10% sugar solution at 28˚C and 80% humidity with a 12-h light/dark cycle. Sterile cotton, filter paper, and sterilized nets were used to maintain the cages as sterilely as possible. For experiments utilizing aseptic mosquitoes, female mosquitoes were maintained on a 10% sucrose solution with 20 units of penicillin and 20 mg of streptomycin per mL (15070–063, Thermo Fisher Scientific) for 5 days. The effectiveness of the antimicrobial treatment was confirmed by colony-forming unit assays prior to blood feeding. Briefly, two mosquitoes from each cup were collected and decontaminated in 75% ethanol and rinsed in sterile PBS. Midguts were then dissected under aseptic conditions, transferred to a microcentrifuge tube containing 200 μl of sterile PBS and macerated for 30 sec. The mixture was then plated on LB agar and kept at room temperature for 48 h. Mosquitoes were confirmed as aseptic if no colonies were observed.

## Membrane blood feeding and detection of mosquito infection rate

The mosquitoes were starved for 24 hr to allow the antibiotics to be metabolized prior to membrane feeding. *Csp_BJ* was grown overnight in liquid LB at 30˚C. The overnight culture was diluted using fresh LB to OD600 = 1.0, which equals a concentration of approximately $10^8$ CFU/ml, and was further diluted to $10^4$–$10^6$ CFU/ml. Fresh human blood from healthy donors was placed in heparin-coated tubes (367884, BD Vacutainer) and centrifuged at $1,000 \times g$ and 4˚C for 10 min to separate plasma from blood cells. The plasma was heat-inactivated at 55˚C for 60 min. The separated blood cells were washed three times with PBS to remove the anticoagulant. The blood cells were then resuspended in heat-inactivated plasma. Mixtures containing human blood (50% v/v), different dilutions of *Csp_BJ* bacterial suspension (25% v/v) and supernatant from DENV- or ZIKV-infected Vero cells (25% v/v) were used to feed antibiotic-treated *A. aegypti* via a membrane feeding system (6W1, Hemotek). Mixtures containing human blood (50% v/v), fresh LB broth (25% v/v) and supernatant from DENV- or ZIKV-infected Vero cells (25% v/v) served as negative controls. Mixtures containing human blood (50% v/v), $10^6$ CFU/ml *Csp_BJ* bacterial suspension (25% v/v) and fresh VP-SFM medium (25% v/v) served as uninfected controls. We used approximately 1 ml of liquid mixture per feeder and allowed the mosquitoes to feed in a dark environment at 28˚C and 80% humidity for 30 min. A total of 20–40 mosquitoes were allowed to feed in each group (numbers may differ for certain experiments). Mosquitoes were then anesthetized by cooling them in a 4˚C refrigerator. Once the mosquitoes were fully anesthetized, they were transferred to a petri dish

embedded in an ice bucket. Mosquitoes were sorted by looking carefully at the abdomens for any sign of a blood meal, and those that had not fed were euthanized and discarded. Fully engorged mosquitoes were put back into cardboard cups, provided a sugar meal (a piece of cotton soaked in 10% sucrose), and incubated under standard rearing conditions for 8 days. Every day, the mosquitoes were monitored for the mortality rate. At Day 8, dead mosquitoes were picked out and were not counted. Mosquitoes that were alive were subsequently euthanized and counted as total number of mosquitoes. Each whole mosquito body was subjected to RNA extraction, reverse transcription and qPCR. When detecting DENV or ZIKV by qPCR, mosquito samples above the limit of detection were considered infected and were counted. DENV/ZIKV infection rate % = number of infected mosquitoes/number of total collected mosquitoes × 100%.

### Identifying virucidal components of *Csp_BJ* culture suspension

For mosquito feeding experiments, aseptic mosquitoes were prepared and maintained as previously described. **(a)** Either the bacterial cell lysate or the culture supernatant (25% v/v) was mixed with human blood (50% v/v) and DENV or ZIKV virus supernatant (25% v/v) for membrane blood feeding of antibiotic-treated *A. aegypti*. Fresh LB broth or the original *Csp_BJ* culturing suspension (25% v/v) mixed with human blood (50% v/v) and virus supernatant (25% v/v) served as negative or positive controls, respectively. The original *Csp_BJ* culturing suspension (25% v/v) mixed with human blood (50% v/v) and fresh VP-SFM (25% v/v) served as an uninfected control for viral genome detection in mosquitoes. **(b)** Either the upper retentate or the lower liquid filtrate (25% v/v) was mixed with human blood (50% v/v) and the virus supernatant (25% v/v) for membrane blood feeding of antibiotic-treated *A. aegypti*. Fresh LB broth or the unseparated cell-free culture supernatant (25% v/v) mixed with human blood (50% v/v) and virus supernatant (25% v/v) served as negative or positive controls, respectively. The unseparated cell-free culture supernatant (25% v/v) mixed with human blood (50% v/v) and fresh VP-SFM (25% v/v) served as an uninfected control for viral genome detection in mosquitoes. Mosquito infectivity was determined by RT–qPCR at 8 days post-blood meal. The final DENV or ZIKV titer was $1 \times 10^5$ PFU/mL for oral infection.

For plaque assays: **(a)** Either the bacterial lysates or the culture supernatant (50% v/v) mixed with 50 PFU of DENV or ZIKV in VP-SFM medium (50% v/v) was incubated for 1 hr before being used for the infection of Vero cell monolayers. Fresh LB broth (50% v/v) mixed with virus-containing medium (50% v/v) served as a negative control. Either the bacterial lysates or the culture supernatant (50% v/v) mixed with fresh VP-SFM medium (50% v/v) served as uninfected controls. **(b)** Either the retentate or the filtrate (50% v/v) was mixed with 50 PFU of DENV or ZIKV in VP-SFM medium (50% v/v) and incubated for 1 hr before inoculation into Vero cell monolayers. Fresh LB broth (50% v/v) mixed with virus in VP-SFM medium (50% v/v) served as a negative control. The culture supernatant (50% v/v) mixed with virus in VP-SFM medium (50% v/v) served as a positive control. Either the retentate or the filtrate (50% v/v) mixed with fresh VP-SFM medium (50% v/v) served as uninfected controls. Plaque assays were performed as previously described. No plaques were observed in uninfected control wells. Inhibition ratios were calculated as follows: Inhibition ratio% = 100%—number of plaques in sample well/number of plaques in negative control well × 100%.

### Viral genome quantitation by RT–qPCR

Total RNA was isolated either from homogenized mosquitoes or infected cell supernatant using a Multisource RNA Miniprep Kit (AP-MN-MS-RNA-250, Axygen) according to the manufacturer's protocol. Total RNA was reverse transcribed to cDNA using an iScript cDNA

Synthesis Kit (1708890, Bio–Rad) in a mixture of 1 μl of iScript Mix and 4 μl of RNA. Reverse transcription was performed at 25˚C for 5 min, 46˚C for 60 min and 95˚C for 1 min. Viral genomes were then quantified by qPCR using iTaq Universal SYBR Green Supermix (1725121, Bio–Rad) according to the manufacturer's protocol using 1 μl of cDNA for each reaction. RT–qPCR was performed on a Bio–Rad CFX-96 Touch Real-Time Detection System. The RT–qPCR cycle was set to 95˚C for 2 min, 40 cycles of 95˚C for 10 sec and 60˚C for 30 sec, detection of melting curve and finally at 25˚C for 2 min. Primer sequences are shown in S1 Table.

## Monitoring mosquito mortality rate after sugar/blood meals

Aseptic mosquitoes, human blood and a serial dilution of *Csp_BJ* culture suspension were prepared as described above. A mixture containing 1% sugar solution or human blood (50% v/v) and different dilutions of *Csp_BJ* bacterial suspension (50% v/v) was used to feed antibiotic-treated *A. aegypti* via a membrane feeding system (6W1, Hemotek). Mixtures containing 1% sugar solution or human blood (50% v/v) and fresh LB broth (50% v/v) served as negative controls. Fully engorged female mosquitoes were transferred into new containers and maintained under standard conditions. Mosquito mortality was observed over 4 days.

## Mass spectrometry

*Chromobacterium sp*. Beijing was pelleted from the culture stock by centrifugation at 4000 × g and 4˚C for 30 minutes (Avanti J-26XP, Beckman). The pellet was then washed three times with cold PBS and suspended in cold VP-SFM medium. After incubation for 2 hr at 37˚C, the bacteria were pelleted and removed by centrifugation and filtration using a 0.22 μm filter unit (SLGP033RS, Millipore). The protein component of the bacterial supernatant was concentrated using an Ultra-15 centrifugal filter concentrator (UFC900396, Millipore) at 4000 rpm and 4˚C for 15 min repeatedly until the desired concentration was reached, and the proteins were then subjected to SDS–PAGE. Fresh VP-SFM medium served as a negative control. The SDS–PAGE gel was excised with a clean razor blade on a glass plate, and the pieces were placed into a 1.5 ml low-binding microcentrifuge tube with destaining buffer (50% acetonitrile, 50% 100 mM EPPS, pH 8.5 in ultrapure water). Gel pieces were dehydrated using acetonitrile, and in-gel enzymatic digestion was performed using trypsin at 37˚C overnight with shaking. Peptide extraction was performed using a solution of 1% formic acid and 75% acetonitrile in ultrapure water. Samples were vacuum centrifuged to dryness and reconstituted in a 5% formic acid and 5% acetonitrile buffer before analysis by liquid chromatography–mass spectrometry (LC–MS) at the Protein Chemistry Technology Core, Tsinghua University. The MS readouts were searched against the protein sequence database of *Chromobacterium spp*. within the Uni-Prot Database using Mascot software. The secreted proteins with a score ≥ 800 were included in the subsequent investigation.

## Protein expression and purification

The genes identified from mass spectrometry were amplified from *Csp_BJ* cDNA and cloned into the pET-28a (+) expression vector. Briefly, DNA extraction of *Csp_BJ* was performed using the E.Z.N.A. Bacterial DNA Kit (D3350, Omega Bio-Tek) according to the manufacturer's protocol. The genes identified from mass spectrometry were then amplified with the primers described in S1 Table. The amplification was performed using 2×TransTaq-T PCR SuperMix (AS122, Transgen) with a PCR cycle of 94˚C for 5 min, 35 cycles of 94˚C for 30 sec, 54˚C for 30 sec and 72˚C for 90 sec, and finally 72˚C for 10 min. DNA fragments were recovered and purified with an E.Z.N.A. Gel Extraction Kit (D2500, Omega Bio-Tek) according to the manufacturer's protocol. The recovered DNA fragments were then cloned into the pET-

28a (+) expression vector using the pEASY-Basic Seamless Cloning and Assembly Kit (CU201, Transgen) according to the manufacturer's protocol.

Expression of recombinant proteins was induced in the *E. coli* BL21 DE3 strain using 200 mM IPTG for protein expression in the soluble form and 500 mM IPTG for inclusion bodies. The proteins were induced by 200 mM IPTG overnight at 16°C to generate soluble forms and purified with TALON metal affinity resin (635501, Clontech). The proteins were eluted with 250 mM imidazole and subsequently dialyzed in PBS buffer (pH 7.4). The inclusion bodies were washed three times with lysis buffer (50 mM Tris, 150 mM NaCl, 5 mM $CaCl_2$, 5% Triton X-100 and 1 mM DTT) and once with 2 M urea, dissolved in 8 M urea and dialyzed overnight in renaturation buffer. Endotoxin was removed (L00338, GenScript) before the protein concentration was measured using a Bradford assay (500–0006, Bio–Rad), and the protein purity was analyzed with SDS–PAGE. The purified *Cb*AE-1 and *Cb*AE-2 proteins were identified by liquid chromatography–mass spectrometry (LC–MS) at the Protein Chemistry Technology Core, Tsinghua University as described in the Mass spectrometry section.

The mutated lipase *Cb*AE-1-S187G was generated using a Q5 Site-Directed Mutagenesis Kit (E0554S, NEB) according to the manufacturer's protocol and purified using the same procedures as described for *Cb*AEs. The primers used for generating *Cb*AE-1-S187G are listed in S1 Table.

A truncated form of *Cb*AE-1 which only consists of its GDSL lipase domain (181AA-339AA) was cloned and purified using the same procedures as above. The primers used for generating *Cb*AE-1-truncated are listed in S1 Table.

## Plaque reduction neutralization tests (PRNTs)

Vero cells were seeded at ~$4\times10^5$ cells per well in 6-well plates and then incubated at 37°C overnight before reaching 80–90% confluence. Virus stocks were diluted to 50 plaque-forming units (PFU) per mL and incubated untreated or with a serial dilution of the *Cb*AEs in five-fold steps at 37°C for 1 hr before being added onto Vero cell monolayers for 2 hr of infection. The same amount of GFP expressed using the same vector and purified using the same procedure as *Cb*AEs in the replacement of *Cb*AEs in the procedure served as the negative control. A total of 1 μg of purified recombinant protein (Fig 3C and 3D) or 10 μg of *Cb*AE-1 or *Cb*AE-2 (Fig 3F and 3G) (50% v/v) mixed with fresh VP-SFM medium (50% v/v) served as the uninfected control. Cell monolayers were washed once with PBS and covered with a 1% agarose overlay diluted in DMEM with 2% FBS. After 4–5 dpi (DENV and ZIKV) or 2–3 dpi (JEV, YFV, SINV and HSV), Vero cell monolayers were fixed and stained with 0.8% crystal violet, and the number of PFU per mL was determined. No plaques should be observed in the uninfected controls. Inhibition ratios are calculated as follows: inhibition ratio% = 100%—number of plaques in sample well/number of plaques in negative control well × 100%. The concentration of each protein necessary to inhibit virus infection by 50% ($IC_{50}$) was calculated by comparison with the untreated cells using the dose–response-inhibition model with the 5-parameter Hill slope equation in GraphPad Prism 8.0 (GraphPad Software, USA).

For PRNT tests including $PLA_2$s from other origins, a $PLA_2$ from *A. mellifera* honeybee venom (P9279, Sigma-Aldrich) and a $PLA_2$ from bovine pancreas (P6534, Sigma-Aldrich) were purchased. *A. mellifera* $PLA_2$ served as positive controls; GFP expressed using the same vector and purified using the same procedure as *Cb*AEs served as negative controls.

## SARS-CoV-2 pseudovirus production and neutralization assays

SARS-CoV-2 pseudoviruses were purchased from GenScript, and neutralization activity was measured using the HEK-293T-ACE2 cell line. Briefly, pseudovirus titres were measured by

luciferase activity in relative light units (RLUs) (Bright-Glo Luciferase Assay System, Promega Biosciences, California, USA). Neutralization assays were performed by adding 100 $TCID_{50}$ (median tissue culture infectious dose) of pseudovirus into 10 serial 1:3 dilutions of *Cb*AE-1 or *Cb*AE-2 starting from 50 μg/ml, following incubation at 37˚C for 1 hr and addition of GhostX4/R5 cells. Neutralizing activity was measured by the reduction in luciferase activity compared to that in the controls. The fifty percent inhibitory concentration ($IC_{50}$) was calculated using the dose-response-inhibition model with the 5-parameter Hill slope equation in GraphPad Prism 8.0 (GraphPad Software, USA). Vesicular stomatitis virus G protein (VSV-G) pseudotyped lentiviruses expressing human ACE2 were produced by transient co-transfection of pMD2G (Addgene #12259) and psPAX2 (Addgene #12260) plasmids and the transfer vector pLVX-ACE2Flag-IRES-Puro with VigoFect DNA transfection reagent (Vigorous) into HEK-293T cells to generate the HEK-293T-ACE2 cells for SARS-CoV-2 pseudovirus infection.

## Pre-treatment virus inhibition assays

For pre-infection treatment, Vero cells were seeded in 24-well plates and allowed to form monolayers. Ten micrograms/mL *Cb*AE-1 or *Cb*AE-2 was added to Vero cell monolayers at 37˚C for 1 hr, and then DENV or ZIKV (0.1 MOI) was added and incubated for another hour at 37˚C. The same amount of GFP expressed using the same vector and purified using the same procedure as *Cb*AEs in replacement of *Cb*AE-1 or *Cb*AE-2 in the procedure served as negative controls. After infection, cell monolayers were washed once with PBS buffer, fresh VP-SFM medium was added, and the cells were incubated at 37˚C for 48 hr before the supernatant was collected for RT–qPCR quantitation of the viral genome.

## Lipase activity assay

The lipase activity of *Cb*AE-1, *Cb*AE-2 and *Cb*AE-1-S187G was measured with a plate assay as previously described [40]. Briefly, 20 μg of *Cb*AE-1, *Cb*AE-2, *Cb*AE-1-S187G or BSA was spot inoculated onto a 2% agar plate with 1% egg yolk and incubated for 24 hr at 37˚C. BSA served as a negative control. Phospholipase activity was indicated by the diameter of the lytic halo around each well.

## Viral RNA exposure assay

DENV ($1 \times 10^4$ PFU) or ZIKV ($1 \times 10^4$ PFU) was incubated with different concentrations of the *Cb*AEs or GFP in a total volume of 1 mL for 1 hr at 37˚C. The same amount of GFP expressed using the same vector and purified using the same procedure as *Cb*AEs in replacement of *Cb*AEs in the procedure served as negative controls. The mixtures were then treated with 1 μL of RNase A (GE101-01, Transgen) and incubated for 1 hr at 37˚C. Viral RNA was extracted using a Multisource RNA Miniprep Kit (AP-MN-MS-RNA-250, Axygen) according to the manufacturer's protocol. RNase A was neutralized in the first step of the RNA extraction procedure by adding Buffer R-I to the mixture. RNA degradation was evaluated by RT–qPCR using the same kits and protocols as mentioned in the viral genome quantification section.

## Transmission electron microscopy

ZIKV viral particles were purified as described previously [41]. Briefly, virus stocks were pelleted in 8% w/v PEG 8000 at 10,000 ×g for 1 hr, purified by 24% w/v sucrose cushion for 2 hr at 175,000 ×g (Beckman SW41 Ti rotor) and separated in a 10–35% potassium tartrate-glycerol gradient for 2 hr at 175,000 ×g (Beckman SW41 Ti rotor). Purified viral particles were suspended in 10 μg/mL BSA solution or 10 μg/mL *Cb*AE-1 solution and incubated for 1 hr at

room temperature. The samples were then applied to a carbon grid, washed 3 times with water and negatively stained with 1% w/v uranyl acetate. The images were acquired by a Hitachi H-7650B TEM microscope at 80.0 kV.

## Cytotoxicity assay

The cytotoxicity of *Cb*AE-1 and *Cb*AE-2 was tested on Vero cells and C6/36 cells. Briefly, the cell viability was measured by the MTT [3-(4,5-dimethylthiazol-2-yl)-2,5-diphenyl tetrazolium bromide] (M8180, Solarbio) method. Confluent cell monolayers contained in 48-well plates were exposed to different concentrations of the *Cb*AEs for 24 hr at 37˚C. Then, a final concentration of 0.5 mg/ml MTT was added to each well. After 4 hr of incubation at 37˚C, the supernatant was removed, and 250 μl of dimethyl sulfoxide (DMSO) was added to each well to solubilize the formazan crystals. After shaking for 10 min, the absorbance was measured at 490 nm. The concentration of each protein necessary to reduce cell viability by 50% ($CC_{50}$) was calculated by comparison with the untreated cells using a sigmoidal nonlinear regression function to fit the dose–response curve in GraphPad Prism 8.0 (GraphPad Software, USA).

## Quantification and statistical analysis

The animals were randomly allocated into different groups. Mosquitoes that died before the measurement were excluded from the analysis. The investigators were not blinded to the allocation during the experiments or to the outcome assessment. No statistical methods were used to predetermine the sample size. Descriptive statistics are provided in the figure legends. Given the nature of the experiments and the types of samples, the differences between continuous variables were assessed using a nonparametric Mann–Whitney test. Differences in mosquito infection rates were analyzed using Fisher's exact test. The survival rates of the infected mice were statistically analyzed using the log-rank (Mantel-Cox) test. *P* values were adjusted using Dunnett's test or the Benjamini–Hochberg procedure to account for multiple comparisons. All analyses were performed using GraphPad Prism statistical software (GraphPad Software, San Diego California USA).

## Supporting information

**S1 Fig. Sequence comparison of *Cb*AE-1 and its homolog in *Csp_BJ*.** (A) Conserved domains of *Cb*AE-1 and *Cb*AE-2 protein sequences were analyzed using a Simple Modular Architecture Research Tool (SMART). (B) Sequence comparison of *Cb*AE-1 and *Cb*AE-2 was performed using the Basic Local Alignment Search Tool (BLAST) on the NCBI website with the program "Needleman-Wunsch alignment of two sequences".
(TIF)

**S2 Fig. The virucidal activity of *Cb*AEs and PLA₂s against DENV and ZIKV.** (A, B) Inhibition curves of *Cb*AE-1, *Cb*AE-2, *A.mellifera* PLA$_2$ and bovine pancreas PLA2 against DENV (A) and ZIKV (B). Standard plaque reduction neutralization tests (PRNTs) were performed. Serial concentrations of proteins were mixed with 50 PFU of DENV or ZIKV in VP-SFM medium to perform standard plaque reduction neutralization tests (PRNTs). GFP was used as negative controls.
(TIF)

**S3 Fig. The virucidal activity of *Cb*AE-1 and *Cb*AE-1-truncated against DENV and ZIKV.** (A, B) Inhibition curves of *Cb*AE-1 and *Cb*AE-1-truncated against DENV (A) and ZIKV (B). Standard plaque reduction neutralization tests (PRNTs) were performed. Serial concentrations of *Cb*AE-1 and *Cb*AE-1-truncated were mixed with 50 PFU of DENV or ZIKV in VP-SFM

medium to perform standard plaque reduction neutralization tests (PRNTs). GFP was used as negative controls.
(TIF)

**S4 Fig. The virucidal activity of *Cb*AEs against HSV-1 and SARS-CoV-2 pseudovirus.** (A, B) Inhibition curves of *Cb*AE-1 and *Cb*AE-2 against HSV-1 (A) and SARS-CoV-2 pseudovirus (B). Standard plaque reduction neutralization tests (PRNTs) (A) or luciferase-based neutralization assays (B) was performed. GFP was used as a negative control.
(TIF)

**S5 Fig. Toxicity evaluation of the *Cb*AEs in Vero cells and C6/36 cells.** (A, B) Cytotoxicity of *Cb*AEs to Vero cells (A) and C6/36 cells (B) was measured by MTT assays.
(TIF)

**S6 Fig. The virucidal and entomopathogenic properties of *Cb*AE-1 in mosquitoes not treated by antibiotics.** (A-F) The virucidal activity of *Cb*AE-1 against DENV (A-C) and ZIKV (D-F) infection in *A. aegypti* is dependent on its lipase activity. (A, D) Schematic representation of the study design. A mixture containing human blood (50% v/v), 10 μg *Cb*AE-1 or *Cb*AE-1-S187G, and supernatant from DENV- (A) or ZIKV- (D) infected Vero cells was used to feed untreated *A. aegypti* via a membrane blood feeding system. Mosquito infectivity was determined by RT–qPCR at 8 days post blood meal. The final DENV or ZIKV titre was $1 \times 10^5$ PFU/mL for oral infection. (B, E) Mosquito infection rate after oral supplementation of *Cb*AE-1 or *Cb*AE-1-S187G with DENV (B) or ZIKV (E). (C, F) Mosquito viral load after oral supplementation of *Cb*AE-1 or *Cb*AE-1-S187G with DENV (C) or ZIKV (F). (G-I) The entomopathogenic activity of *Cb*AE-1 in *A. aegypti* is dependent on its lipase activity. (G) Schematic representation of the study design. A mixture containing 1% sugar solution (H) or human blood (I) (50% v/v) and 10 μg *Cb*AE-1 or *Cb*AE-1-S187G (50% v/v) was used to feed untreated *A. aegypti* via a membrane blood feeding system. Mosquito mortality was observed over 4 days. (B, C, E, F, H, I) The number of infected mosquitoes relative to total mosquitoes is shown at the top of each column (B, E). Differences in the infectivity ratio were compared using Fisher's exact test (B, E). A nonparametric Mann–Whitney test was used for the statistical analysis (C, F). The survival rates of mosquitoes were plotted using a Kaplan–Meier curve and were statistically analyzed using the log-rank (Mantel-Cox) test (H, I). *P* values were adjusted using the Benjamini–Hochberg procedure (B, E, H, I) or Dunnett's test (C, F) to account for multiple comparisons. The *P* value represents a comparison between the control group and the other groups. $^*P < 0.05$, $^{***}P < 0.001$, $^{****}P < 0.0001$, ns, not significant. The limit of detection is illustrated by dotted lines (C, F). Experiments consisted of at least three biological replicates with similar results.
(TIF)

**S1 Table. Primers for SYBR Green RT-qPCR and gene cloning.**
(DOCX)

## Acknowledgments

We thank the core facilities of the Center for Life Sciences and Center of Biomedical Analysis for technical assistance (Tsinghua University).

## Author Contributions

**Conceptualization:** Xi Yu, Gong Cheng.

**Data curation:** Xi Yu, Liangqin Tong, Liming Zhang, Yun Yang, Xiaoping Xiao, Yibin Zhu.

**Formal analysis:** Xi Yu.

**Funding acquisition:** Yibin Zhu, Gong Cheng.

**Investigation:** Xi Yu, Gong Cheng.

**Supervision:** Gong Cheng.

**Writing – original draft:** Xi Yu, Gong Cheng.

**Writing – review & editing:** Xi Yu, Penghua Wang, Gong Cheng.

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
