## [Decision Letter · Decision Letter 0]

18 Feb 2022

Dear Dr. Cheng,

Thank you very much for submitting your manuscript "Lipases secreted by a gut bacterium inhibit arbovirus transmission in mosquitoes" for consideration at PLOS Pathogens. As with all papers reviewed by the journal, your manuscript was reviewed by members of the editorial board and by several independent reviewers. In light of the reviews (below this email), we would like to invite the resubmission of a significantly-revised version that takes into account the reviewers' comments.

Each of the reviewers had significant concerns about the manuscript. One of the reviewers mentioned that there were major deficiencies. A revised manuscript will be considered only after all the comments from the reviewers are addressed.

We cannot make any decision about publication until we have seen the revised manuscript and your response to the reviewers' comments. Your revised manuscript is also likely to be sent to reviewers for further evaluation.

Sincerely,

Richard J. Kuhn, PhD

Associate Editor

PLOS Pathogens

Sara Cherry

Section Editor

PLOS Pathogens

Kasturi Haldar

Editor-in-Chief

PLOS Pathogens

orcid.org/0000-0001-5065-158X

Michael Malim

Editor-in-Chief

PLOS Pathogens

orcid.org/0000-0002-7699-2064

Each of the reviewers had significant concerns about the manuscript. One of the reviewers mentioned that there were major deficiencies. A revised manuscript will be considered only after all the comments from the reviewers are addressed.

Reviewer's Responses to Questions

**Part I - Summary**

Reviewer #1: The manuscript by Yu et al. described the antiviral activities of two lipases secreted by a gut bacterium from mosquito vector. The lipases CbAE-1 and CbAE-2 shows broad spectrum antiviral activities against various arboviruses and also have entomopathogenic effect to mosquitoes. The authors further suggested the lipase activity directly disrupted the viral envelope and also critical for their entomopathogenic activity. Overall, the experimental procedures are rigorous; the results are novel and convincing. The manuscript is well organized and clearly written. I would suggest the authors to address the following questions to improve the manuscript:

Reviewer #2: This manuscript describes the isolation of two lipases from a Chromobacterium species that demonstrates virucidal activity. The study represents an impressive amount of work that consistently follows through to identify the mechanism of action of these enzymes and the cause for the virucidal activity. Therefore, this study will be of significant interest to the readership of the journal and to the field. Prior to publication, some concerns need to be addressed - primarily the use of necessary controls are either missing or not well-described in the manuscript.

Reviewer #3: The manuscript describes in vitro and in vivo experiments to test the effect of two bacterial lipases against selected arboviruses. The authors claim to have identified two lipases with virucidal and entomopathogenic effects in mosquitoes. It is difficult to evaluate the study results and claims properly and determine the overall significance of this study. The authors did not indicate what type of negative and positive control they used for most experiments and did not provide enough information on most experimental procedures. In addition, they did not describe the results in sufficient detail and compared them with previous studies. Overall, the manuscript is not well organized and adequately developed.

**Part II – Major Issues: Key Experiments Required for Acceptance**

Reviewer #1: 1) CbAE-1 and CbAE-2 have 52% identity, but CbAE-1 seems have much stronger antiviral activities. What makes this difference?

2) Since Chromobacterium bacterium and CbAE-1 and CbAE-2 are harmful to mosquitoes, can you estimate how much Chromobacterium bacterium are inside the mosquito gut in the real world.

3) The molecular weights of CbAE-1 and CbAE-2 is around 50kDa which means they are not easy to produce. Also, they will face the problems of salability and immunogenicity. Did you take any effort to make some truncated forms to get the minimum active form of these proteins? For example, did the lipase domain alone show any activity?

4) The data suggest that CbAEs have antiviral activities against DENV, ZIKV, JEV, YFV and SINV. The last one is not a flavivirus, could it have different mechanisms?

5) Have you ever test whether CbAEs have antiviral activities against other enveloped viruses except for these mosquito-borne viruses? If so, this would be interesting because CbAEs have the potential as broad-spectrum antiviral drug candidates.

6) Considering CbAE-1/-2 disrupt the lipid associate membranes, have you ever tested the cytotoxic effect of these proteins to mammalian cells?

Reviewer #2: Cell lysate – how was it processed? Lots of effectors may be membrane bound and may not get into the midgut.

Fig 1B/E – how was “dengue infection rate %” calculated?

Fig 1C/F – viral titer in whole mosquitoes?

Figure 1G-I – “bacterial suspension” – does it include supernatant also, or bacteria were spun down and resuspended? In other words, were secreted metabolites included in this suspension?

Figure 1H and 1I – unclear from text or legend as to what the difference between these are?

Fig2A/B – When the authors carried out the incubation expts with the supernatant, upper retentate and lower filtrate with the 50PFU of virus, was a time zero sample quantified. This would be important to demonstrate that the activity of an enzyme rather than some non-specific inhibition is occurring.

Fig3F and 3G – was a positive control protein such as a known PLA2 and negative control such as BSA or GFP used? These would be important for comparison if generic lipase activity is the key effector here. Else, is it the specific activity of CbAE-1/2 compared to other lipases and if so what is the hypothesis? Also, can the authors include a cytotoxicity assay for these protein IC50 curves since the protein is not removed following incubation with the virus?

Materials and methods have 'mice' mentioned, but these data are not discussed in the paper.

Fig 3E - Protein expression and purification – how did the authors validate the identity of the purified proteins CbAE-1/2?

Fig 3F/G – were control proteins (positive control for lipase activity) and a negative control (non-lipase such as BSA or GFP) used in these experiments? If not, they will need to be included.

Fig4F – can the authors provide a larger magnification of a select number of control and treated virus particles? It is quite difficult to see the structural differences and a zoomed in image will help. 

Reviewer #3: The manuscript is not formatted as per journal guidelines. The following sections are not indicated in the main text: Author Summary, Introduction, Results, Discussion.

Missing experimental procedures that should be added to the material and methods:

1. Methods used to identify the Chromobacter Beijin from the gut of the mosquito.

2. Sequencing methods and sequencing comparison methods.

3. No experimental method corresponding to Figure 1. G to I is described in this section.

4. Method used to produce the mutated lipase

5. Plaque assay and mosquito feeding experiments of Figure 2

In the material and methods section, there is a mention of mice experiments (Lines 358-359), but no detailed experimental procedures are provided and there is no data linked to these experiments in the manuscript.

Negative and positive control are not indicated for experimental procedures: Lanes 76-84; 85-87; 94-106; 159-167; 429

**Part III – Minor Issues: Editorial and Data Presentation Modifications**

Reviewer #1: (No Response)

Reviewer #2: Line 90 – “resist” is the wrong word to use

Additional checks of word usage would be good.

Reviewer #3: Main Text

Lines 60-61: this sentence is unclear. What do the authors mean by “secreted bacterial proteins…. that resist infections”?

Lines 70-76: Why did the authors not compare the sequence of the Csp_BJ with the Csp_P?

Lines 76 to 84: A detailed experimental procedure is missing from the material and methods. The infection rate of the positive controls was ~60% for DENV and ~50% for Zika. Why? Also, what was the percent mortality in these assays? Why there is a difference in the total number of mosquitoes between groups?

Line 81: the word “prevalence” does not seem appropriate in this context.

Line 83: this sentence is unclear. The authors should explain how their results imply a “close relationship” between Csp_BJ and Csp_P strains.

Lines 85-87: What was used as negative control? Why was 1% sugar solution used in this experiment? How was the bacterial concentration range determined? Why there is a difference in the total number of mosquitoes between groups? Was the mosquito mortality observed in this assay comparable with the one observed in the previous feeding assay? There is a ~ 10% mortality in the blood fed control and no mortality in the controls fed with sugar water. Why?

Line 90: what do the authors mean by “Csp_BJ resists viral infection in mosquitoes”? The sentence is unclear. The word “resists” does not seem appropriate.

Lines 92-94: this sentence is unclear, what does “differential fragmentation” refer to?

Lines 94-106: In this set of experiments, there is a difference in the total number of mosquitoes between groups and between experiments (DENV vs. ZIKV)? Why? Also, what was fed to the negative and positive controls group of mosquitoes?

In the Figure 2 legend (Lines 254-255), negative (LB broth) and positive (cell culture supernatant +virus) controls are mentioned. I think that a better negative control would be using the bacterial supernatant, but without the virus. How was the percent inhibition ratio calculated?

Line 100: the word “prevalence” does not seem appropriate.

Lines 115-130: What was the expected size of the expressed proteins? What was used as negative/positive control in the assays of Figure 3-C-D-F-G?

Line 132: The sentence is unclear. The word “resists” does not seem appropriate.

Lines 138-139: How was the S187G mutation produced? There is no information in the material and methods.

Lines 144-145: Why was the BSA used as control? I think an unrelated protein expressed using the same vector and purified using the same procedure as the two lipases should be a more appropriate negative control.

Lines 159-167: what was used as a control in these experiments?

Line 162: the word “catalytic” does not seem appropriate.

Lines 169-173: There is a difference in the total number of mosquitoes between groups and between experiments (DENV vs. ZIKV)? Why? Also, the infection rate is very low for the BSA control in DENV (33%), why?

Why did the authors used antibiotic-treated mosquitoes to test the lipases? Would the lipases effect on virus infection rate in mosquitoes be the same in the presence of a normal gut microflora?

Material and Methods

Lines 380-384: The blood feeding experiment needs more details. How much blood was used on the Hemoteck? How many mosquitoes were used/experiment? For how long were the mosquitoes left feeding, and under what conditions (temperature, humidity, light)? What method was used to transfer the fully fed mosquitoes into new containers? What was the mixture fed to the negative/positive controls?

Line 388-389: the authors should describe the protocol used to test the mosquitoes to confirm the removal of gut bacteria. How were the mosquitoes processed? How many mosquitoes were used?

Line 397: Was the bacterium pelleted from the culture stock before the washes in PBS? If so, please indicate the centrifuge used (make and model) and centrifugation xg, time and temperature used. Also, if the protein concentration step was done using a centrifuge, please indicate the centrifuge used (make and model) and centrifugation xg, time and temperature used. How were the proteins purified from the SDS-PAGE gel before LC-MS?

Line 404: the gene amplification and cloning procedure needs more details: DNA extraction method, primers used, PCR cycle, cloning procedure.

Line 437: This paragraph needs more details: reaction mix used for dsRNA synthesis, RT-qPCR cycle.

Line 446: How was “CbAE-1-S187G” obtained?

Lines 454-455: How was the RNase A neutralized? What kits were used for RNA extraction and RT-qPCR.

PLOS authors have the option to publish the peer review history of their article (what does this mean?). If published, this will include your full peer review and any attached files.

Reviewer #1: No

Reviewer #2: **Yes: **Rushika Perera

Reviewer #3: No
---

## [Editor Report · Decision Letter 1]

26 Apr 2022

Dear Dr. Cheng,

We are pleased to inform you that your manuscript 'Lipases secreted by a gut bacterium inhibit arbovirus transmission in mosquitoes' has been provisionally accepted for publication in PLOS Pathogens.

Best regards,

Richard J. Kuhn, PhD

Associate Editor

PLOS Pathogens

Sara Cherry

Section Editor

PLOS Pathogens

Kasturi Haldar

Editor-in-Chief

PLOS Pathogens

orcid.org/0000-0001-5065-158X

Michael Malim

Editor-in-Chief

PLOS Pathogens

orcid.org/0000-0002-7699-2064

This is a revised manuscript. The authors responded to critiques from three reviewers and had extensive revisions based on their questions and comments. Several new experiments and controls were done in response to the reviewers. The authors have adequately addressed these numerous concerns and have improved significantly their manuscript.
---

## [Editor Report · Acceptance letter]

16 May 2022

Dear Dr. Cheng,

We are delighted to inform you that your manuscript, "Lipases secreted by a gut bacterium inhibit arbovirus transmission in mosquitoes," has been formally accepted for publication in PLOS Pathogens.

Best regards,

Kasturi Haldar

Editor-in-Chief

PLOS Pathogens

orcid.org/0000-0001-5065-158X

Michael Malim

Editor-in-Chief

PLOS Pathogens

orcid.org/0000-0002-7699-2064